# Effect of Residual Stress Induced by Different Cooling Methods in Heat Treatment on the Fatigue Crack Propagation Behaviour of GH4169 Disc

**DOI:** 10.3390/ma15155228

**Published:** 2022-07-28

**Authors:** Menglong Fan, Chuanyong Chen, Haijun Xuan, Hailong Qin, Mingmin Qu, Songyi Shi, Zhongnan Bi, Weirong Hong

**Affiliations:** 1High-Speed Rotating Machinery Laboratory, College of Energy Engineering, Zhejiang University, Hangzhou 310027, China; 11528007@zju.edu.cn (M.F.); marine@zju.edu.cn (H.X.); hongwr@zju.edu.cn (W.H.); 2Beijing Key Laboratory of Advanced High-Temperature Materials, Central Iron and Steel Research Institute, Beijing 100081, China; hailongqin@126.com (H.Q.); shisybj@163.com (S.S.); bizhongnan21@aliyun.com (Z.B.); 3Beijing GAONA Materials and Technology Co., Ltd., Beijing 100081, China; 4Zhejiang HIRO Aeronautics Technology Co., Ltd., Huzhou 313219, China; mm_hiroma@zju.edu.cn

**Keywords:** residual stress, heat treatment, crack propagation, surface replica method, spin test

## Abstract

In this study, the effects of residual stress induced by three different cooling methods during heat treatment on the crack propagation behaviour of the GH4169 disc were investigated. Different levels of stress fields were induced to the specially designed discs by using air cooling (AC), air jetting cooling (AJC) and water quenching (WQ) methods and were quantitated by numerical simulation. These discs were then subjected to prefabricated cracking, and crack propagation tests were conducted on a spin tester with two load spectrums. Crack growth behaviour was depicted via the surface replica technique and fracture morphology. Regarding the linear superposition of residual stress and centrifugal force, the crack propagation behaviour of different discs was simulated using the FRANC3D software. AJC and WQ introduced compressive residual stress (−259 MPa and −109 MPa, respectively) into the disc compared with the AC method (about −1.5 MPa). The AJC method increases the crack propagation life of the disc by introducing residual compressive stress into the area near the surface of the central hole to inhibit the opening of the crack surface. When the fatigue load was low, this inhibition effect was more significant.

## 1. Introduction

The nickel-based superalloy GH4169 is a key material for manufacturing aero-engine turbine discs because of its attractive high-temperature mechanical properties, such as exceptionally high-temperature strength, good resistance to oxidation and fatigue [1,2]. Turbine discs suffer from low-cycle fatigue (LCF) failure, one of the most serious and costly problems in the service process [3]. Usually, an aero-engine disc is designed using the traditional safe-life method, which refers to the design that makes the fatigue failure probability of a load-bearing structure very small without inspection and maintenance within the specified life period [4]. However, the main shortcoming of the safe-life method in the life management of fracture critical components is the failure to explicitly consider the existence of imperfections that come from material processing, manufacturing and usage. These rare defects can grow to a critical size faster than safe-life predicted cycles and lead to severe incidents [5,6,7,8]. Fatigue crack growth has been identified as one of the most important causes affecting the long-term structural integrity of mechanical components and engineering structures subjected to cyclic loading [9]. Based on the above-mentioned reasons, improving aero-engine disc fatigue resistance is the main consideration in disc design and manufacturing.

Residual stresses are self-balanced and nonhomogeneous stresses generated during different manufacturing and heat treatment processes of engineering components [10]. One can take advantage of these stresses, purposely causing compressive residual stresses into the surface layer of components to improve fatigue life [11,12]. Contrarily, during operation, the residual tensile stress that appears in the material will facilitate the initiation and propagation of cracks [13,14], which should be avoided as far as possible. A wealth of research has established that residual stresses can alter the mechanical and functional performance of different engineering systems [15,16,17,18,19]. Various surface treatment technologies, such as laser shock peening (SP) [20,21], low plasticity burnishing (LPB) [22] and hole extrusion strengthening (HES) [23], are utilised to induce residual stress. Amongst them, SP and HES are the most traditional techniques for producing near-surface and hole wall residual stress, respectively. The SP method uses high-speed projectile flow (20–150 m/s) to produce elastoplastic deformation of surface material and form a region of a beneficial residual compressive stress field, given that rebounding of the elastic deformation squeezes the plastic region [24]. Especially in recent years, simulations and experimental investigations of surface quality and residual stresses in shot peening were frequently studied [25,26,27,28]. HES uses a certain amount of interference mandrel to squeeze the inner surface of the hole structure to produce controllable radial plastic deformation and introduces hundreds of MPa residual compressive stress to reduce the stress level of the hole edge and improve the fatigue life [29,30]. However, in some small inner holes and grooves, SP is difficult to reach, resulting in dead corners whose parts are often under concentrated stress. In HES, the interference is difficult to control, and high interference will cause damage to the hole wall, whilst low interference will introduce residual tensile stress into the hole wall, increase the fretting wear between the hole wall and the liner, and reduce the fatigue performance of the connecting hole [31,32]. Some studies have found that the residual stress of GH4169 plate forgings after quenching can reach about 400 MPa [33,34], and this residual stress is difficult to fully release during the subsequent ageing or annealing treatment [35,36]. Hence, a special heat treatment process can be selected in the blank disc stage to make the cooling rate of the inner hole much faster than that of other areas to generate a certain value of residual compressive stress which will improve the resistance of crack initiation and propagation from the inner hole surface.

In this study, three cooling ways, air cooling (AC), air jetting cooling (AJC) and water quenching (WQ), were adopted after the ageing treatment of GH4169 blank to induce different residual stress levels. Then, the detailed residual stress distribution was analysed using the finite element method. A series crack propagation test of five discs was carried out to study the crack propagation feature with initial flaws prefabricated by electrical discharge machining (EDM). The crack growth data was recorded by fluorescence detection and surface replica methods after interval detection, post-test fracture surface morphology and FRANC3D simulation. Lastly, a comparison of measured and predicted crack size evolution curves was conducted, and the effect of residual stress by different cooling ways in heat treatment on the fatigue crack propagation behaviour of GH4169 discs was discussed.

## 2. Disc Design and Residual Stress Introduction

### 2.1. Material Properties and Disc Design

The nickel-based superalloy GH4169 is widely applied in the manufacturing of aero-engine turbine discs. The basic mechanical properties of the GH4169 alloy at 500 °C are given in Table 1. The turbine disc of the aero-engine is primarily subjected to alternating cyclic thermal stress and centrifugal force loads. Some portions of the turbine disc, such as the tenon groove, central hole, bolt hole and fillet, show a multiaxial stress state, making local stress concentrations easier to induce. A relatively simple turbine disc construction disregarding blades was designed, as shown in Figure 1, to investigate the stress distribution in the central hole and consider the convenience of the connection with the spin tester. Eight high-strength screws joined the machined disc to the shaft.

Detailed stress analysis of the disc under several levels of centrifugal load (42,500 rpm, 45,000 rpm, 47,500 rpm and 50,000 rpm) was performed using the finite element simulation software ABAQUS. A 1/12 3D model of the disc ignoring the connection bolt holes was built, as presented in Figure 2. The eight nodes linear hexahedron element (C3D8R) was utilized, and 12,420 meshes were included in the entire model. The mesh was refined in the area where crack propagation passed, as shown in Figure 2a. The rigid body displacement of the whole model was limited by imposing axial displacement constraints on the upper surface of the flange edge and cyclic symmetric constraints on the two circumferential sections of the model, as shown in Figure 2b. Each node in the model includes three degrees of freedom along the coordinate axis. The centrifugal load was applied to the whole model in the form of speed.

For a high-speed rotating disc, the hoop stress was the major component of the Mises stress in the high-stress region, and it was also the dominant driver of crack propagation. As shown in Figure 3, the hoop stress distribution in the central hole region could be characterised by a univariate field near the surface of the central hole. Under 45,000 and 47,500 rpm, material yielding occurred near the inner surface of the central hole due to stress concentration, whilst the stresses in the other parts of the disc were less than the yield stress under both rotational loads. In this condition, in the LCF test, not only would the entire disc not fracture immediately due to insufficient strength, but the crack in the central hole could grow rapidly, facilitating the acquisition of experimental data.

### 2.2. Formation of Residual Stress by AJC

The standard heat treatment process for GH4169 disc forgings is shown in Figure 4. The process mainly included solution treatment, solution cooling, ageing treatment, and ageing cooling. The solution cooling stage was the main stage of residual stress formation. After the material was kept at 980 °C for 1 h, different cooling ways, including AC or WQ, were used to obtain different microstructures and properties. During the cooling process, the internal structure of the material had no time to change, so it was considered that there was no phase change in the cooling process. Therefore, only the relationship between heat conduction and deformation was considered; that is, the heat conduction between the material and environment caused deformation of the material and then changed the distribution of residual stress in the material. The ageing treatment stage included holding at 720 °C and 620 °C for 8 h to stabilise and decrease the residual stress distribution inside the disc. In the subsequent finishing process, the residual stress in the disc would change slightly, which will be mentioned in the subsequent residual stress simulation.

Thermal stress was the main reason for the formation of residual stress in GH4169. During the cooling process, the cooling rate of the material surface was greater than that of the interior, and the material surface would be squeezed by the interior, resulting in a certain amount of residual compressive stress. The special cooling method, AJC, shown in Figure 5, was adopted after solution heat preservation to increase the cooling rate of the central hole surface. The other surface of the workpiece was isolated from the atmosphere by thermal insulation material, and then the inner surface of the central hole of the workpiece was rapidly cooled using high-pressure gas.

### 2.3. Residual Stress Analysis of Different Cooling Ways in Heat Treatment

Fourier’s law is a common model in heat transfer. It holds that in heat conduction, the heat passing through a plane per unit time is proportional to the plane area and temperature gradient along the normal direction of the plane. The heat flux vector *J_T_* (W/m^2^) can be expressed as:(1)$$J_{T}=-k\frac{\partial T}{\partial r}$$
where *k* denotes the thermal conductivity (W/(m^2^ °C)), *T* denotes the temperature, and *r* is the distance. The heat flux vector can also be decomposed into several component representations:(2)$$J_{T}=-k\nabla T=-k\left(\frac{\partial T}{\partial x}i+\frac{\partial T}{\partial y}j+\frac{\partial T}{\partial z}k\right)$$
where *i*, *j*, *k* is the base vector of the Cartesian coordinate system *oxyz*. In heat conduction analysis, the whole system conformed to the first law of thermodynamics, where:(3)ΔU=Q−W
where ΔU denotes the system’s internal energy, Q denotes heat absorbed by the system from the environment, W denotes the work the system does to the environment.

Accordingly, the governing equation of transient heat conduction in finite element calculation can be expressed as follows:(4)$$k\left(\frac{\partial^{2}T}{\partial x^{2}}\right)+k\left(\frac{\partial^{2}T}{\partial y^{2}}\right)+k\left(\frac{\partial^{2}T}{\partial z^{2}}\right)+q=c\rho \frac{\partial T}{\partial t}$$
where q denotes internal heat source intensity, c denotes Specific heat capacity (J/(kg/℃)), ρ denotes material density ($\mathrm{kg}/m^{3}\;$).

The temperature of the material and the fluid medium in contact with it are known, which conforms to the following formula:(5)$${-k\frac{\partial T}{\partial n}|}_{\Gamma }={h\left(T-T_{f}\right)|}_{\Gamma }$$
where *h* denotes Thermal convective coefficient (J/(m^2^s ℃)), $T_{f}$ denotes Fluid medium temperature (°C).

Table 2 lists the relationship between these thermodynamic parameters and temperatures in detail [33]. The thermal convective coefficient of high-pressure jetting air was 1800 J/(m^2^s·°C), whereas it was only 20 J/(m^2^s·°C) in stable air.

Heat treatment cooling was a rapid cooling process, and it could be that the internal structure of the material had no time to change. The change of temperature caused by material heat conduction would affect the distribution of residual stress in the material. The heat generated by element deformation was ignored relative to the temperature change of the material itself. And the four nodes axisymmetric thermally coupled quadrilateral element (CAX4T) was chosen to complete the conversion between material temperature and displacement.

The total strain rate tensor of the material can be decomposed and expressed as the following formula:(6)$$\dot{\varepsilon }={\dot{\varepsilon }}^{e}+{\dot{\varepsilon }}^{p}+{\dot{\varepsilon }}^{T}$$
where $\dot{\varepsilon }$ denotes total strain rate tensor, ${\dot{\varepsilon }}^{e}$*,*
${\dot{\varepsilon }}^{p}$ and ${\dot{\varepsilon }}^{T}$ denotes elastic strain rate tensor, plastic strain rate tensor and thermal strain rate tensor, respectively.

For the thermal expansion of isotropic materials, the thermal strain rate tensor component can be expressed as:(7)$${\dot{\varepsilon }}^{T}=\alpha \dot{T}G$$
where α denotes the coefficient of thermal expansion and *G* denotes the metric tensor.

Based on Hooke’s law, the elastic strain rate tensor can be calculated as follow:(8)$${\dot{\varepsilon }}^{e}=L^{-1}:\dot{\sigma }$$
where $L^{-1}$ denotes elastic stiffness tensor, $\dot{\sigma }$ denotes stress rate tensor.

The plastic strain rate tensor can be expressed as follows:(9)$${\dot{\varepsilon }}^{p}=\frac{3}{2}\dot{p}\frac{e^{\prime }}{\sigma_{e}}$$
where $\dot{p}$ denotes equivalent plastic strain rate, $e^{\prime }$ denotes partial stress tensor and $\sigma_{e}$ denotes equivalent stress.

The fully coupled thermal-stress analysis in commercial finite element software ABAQUS was used to compute the residual stress fields in this paper. We considered that the change of temperature caused by material heat conduction would affect the distribution of residual stress in the material; the heat generated by element deformation was ignored relative to the temperature change of the material itself. The four nodes axisymmetric thermally coupled quadrilateral element (CAX4T) was chosen to complete the conversion between material temperature and displacement.

The whole simulation process was divided into three steps. In the initial step, the basic material parameters and a temperature field of 980 °C were applied to the workpiece. In the second step, the heat transfer coefficients of different surfaces and environments were set to simulate the cooling process after solid solution. Lastly, the stress changes from the rough machined workpiece to the finishing disc were calculated through the setting of the life and death unit. In the last step, the specified elements were removed to simulate the material removal from the rough workpiece. ABAQUS stored the forces exerted by the removed region on the rest of the model at the nodes on the boundary between them. These forces were ramped down to zero during the removal step; therefore, the effect of the removed region on the rest of the model was completely absent only at the end of the removal step. The forces were ramped down gradually to ensure that element removal had a smooth effect on the model. It should be pointed out that the machining shape, size and allowance allocation of discs in actual engineering were much more complex than in this simulation, and multiple processes were needed.

To verify mesh independence, the residual hoop stress fields of three heat treatments using different mesh sizes ranging from 4 mm to 0.3 mm were computed. And the maximum residual hoop stress and the strain energy of the whole model were utilized to evaluate the influence of mesh size on the simulation calculation. As a representative, the maximum residual hoop stress and the strain energy of the whole model of the WQ method are shown in Figure 6. As the grid size interval gradually decreased, the abscissa was represented by the logarithmic coordinate axis. It could be concluded from Figure 6 that when the mesh size was less than 1 mm, the maximum residual hoop stress and the strain energy of the whole model were hardly affected by the mesh size. Consequently, the meshes of 1 mm size were adopted in the finite element calculation of residual stress.

The residual hoop stress distribution of the rough machined workpiece after heat treatment and process with three different cooling methods (AC, AJC, WC) is shown in Figure 7, Figure 8 and Figure 9. The co-authors of the paper, Hailong Qin, measured the residual stress distributions in GH4169 discs quenched with different methods by neutron diffraction [37]. In the ageing heating stage, the material strength decreases gradually, and part of the residual stress (about 33%) is released by plastic deformation. In the ageing insulation stage, the residual stress was mainly released by creep deformation, accounting for about 15%. It is considered that 48% of residual stress will be decreased after ageing treatment. The residual hoop stress in the air-cooled discs was relatively small and almost negligible. After ageing, the thermal stress produced by AC was low, the yield strength of the material was high, and plastic deformation hardly occurred, so the residual stress remained at the same level as before AC [38]. The air-jetting-cooled disc had about 259 MPa residual compressive stress near the central hole, and the residual stress exhibited an almost step distribution along the direction of disc radius. The residual compressive stress was present in the range of 6 mm from the inner diameter of the disc, and the residual tensile stress was present outside the range. The residual compressive stress in the WQ discs was present in the range of 3 mm from the inner diameter of the disc, and the residual tensile stress was present outside the 3 mm from the central hole surface.

## 3. Crack Propagation Test

### 3.1. Prefabrication of Initial Cracks

Two initial flaws were prefabricated at the chamfer of the central hole in the disc by using the EDM technique, as shown in Figure 10. Compared with the inner surface crack of the central hole, the initial prefabricated crack at the chamfer was not only easier to expand due to the larger stress, but also it was easier to observe and measure the morphology and length of the crack in the subsequent test. The flaw size was a semicircle of 0.38 × 0.38 mm. The flaws were oriented perpendicularly to the circumferential direction of the discs so that circumferential stress would be the dominant stress to drive crack propagation during the spin test.

### 3.2. Fatigue Test Plan

A fatigue test was carried out on an HR6DI rotor high-speed rotating tester (Figure 11) in High-Speed Rotating Machinery Laboratory, Zhejiang University. The whole test system mainly included the vacuum test chamber, DC power motor, heating furnace, spindle speed and vibration monitors and disk temperature monitoring systems.

To study the effects of different residual stresses on crack propagation at varying speeds, five discs numbered D01~D05 were processed for comparative tests, as shown in Table 3.

The load spectrum of fatigue is shown in Figure 12. The maximum speed of test *n*_2_ was 47,500 or 45,000 rpm, whilst the minimum speed *n*_1_ was 3150 rpm. The dwell time at the upper speed was 2 s, whilst no stop at the lower speed. All test parameters are listed in Table 4.

### 3.3. Test Results

#### 3.3.1. Disc Life Statistics and Failure Modes

The results of the disc crack propagation test are shown in Table 5.

Every disc was decomposed into four large pieces after fatigue failure, and the screws connecting the disc and the tool shaft were broken in different degrees of shear. A typical fatigue fracture is shown below. The surface of the fracture was smooth and flat. The source zone and part of the propagation zone of fatigue were blackened by high-temperature oxidation. The discs after fatigue failure are shown in Figure 13.

#### 3.3.2. External Surface Morphology Detection of Crack

The initial crack growth rate was slow, and the crack did not extend to the surface at the initial crack growth stage. In the early stages of the experiment, fluorescence detection techniques were used to observe and record the length and direction of crack propagation. The surface replica method [39,40] was adopted to observe and record the length and propagation direction of cracks effectively.

Establishing a reasonable detection period in the LCF test is necessary. A too long detection period will lead to failure, which allows accurate analysis of the law of crack growth; on the contrary, a too short detection period will cause unnecessary economic losses. The test was suspended every 500 cycles during the first 2500 cycles and every 200 cycles after 2500 cycles to observe the crack propagation direction and measure the crack length.

The typical crack morphology and fluorescence detection photos of D01 after 1500 cycles are shown in Figure 14. The crack length was approximately 0.824 mm. The crack propagation direction was almost parallel to the disc diameter direction.

The surface replica method has been proposed as an effective crack detection method in aerospace engine rotor parts. This method is to extrude two colloids, mix them in a mixing tube and smear them on the surface to be detected. After the colloidal mixture is solidified, it is removed and pasted on the slide. In this way, the surface to be detected can be seen clearly by observing the replica plane through an optical microscope. Two surface replica methods are applied broadly: acetate tape replication and silicon rubber-based compound. As the crack propagated through a period of the cycle (approximately after 2000 cycles) in this LCF test, the crack replica method could reproduce the length and shape of the crack. A typical replica of a crack in the D01 disc after 1500, 2000, 2500, 2700, 2900, 3100 and 3300 cycles after crack initiation is shown in Figure 15a–g.

The external surface crack length versus crack propagation life of all five discs recorded using the surface replica method is shown in Figure 16.

#### 3.3.3. Fracture Morphology of the Inner Surface of the Crack

The fracture with the longest crack length regarding each disc was observed under a scanning electron microscope. Owing to the lack of space, only the quantitative fracture analysis of D01 is listed, and the work on D01 can be repeated on the four other discs. There was obvious radiation in the fatigue source region, and there was a certain height difference in the extended region. One fractography direction at approximately 45° of the crack was selected as the follow-up observation direction, as shown in Figure 17.

In the crack propagation region, typical fatigue bands could be observed. The spacing of fatigue bands increased with the crack propagation. The same fatigue band was not continuous, and substantial step faults appeared, indicating that the stress level was high in this area. Subsequently, the fatigue strip spacing at different distances from the fatigue source region was used to analyse the crack growth life of the whole process quantitatively. In the fast fracture zone of the crack, obvious dimple characteristics could be seen clearly. The typical fracture morphology of the crack in different propagation stages observed in the D01 disc is shown in Figure 18.

When measuring the crack growth rate according to the fracture surface, try to select a large number of strips with uniform distribution and clear contour, and measure multiple side-by-side fatigue strips in the same measurement area, then take the average value as the measured data to reduce the error caused by various factors. Meanwhile, two paths shown in Figure 19 of each fatigue fracture were picked to reduce the error caused by the measurement process. The inner surface crack propagation rate versus crack length of all five discs calculated by crack fracture morphology is shown in Figure 20. For the discs with different cooling methods, the difference in crack growth rate on the propagation path near the surface was smaller than that on the central path. And for the same fatigue load, the crack growth rate of the disc treated by the AJC method was clearly lower than by the AC and WQ methods.

## 4. Analysis and Discussion

After different cooling methods during solution treatment, the discs presented diverse residual stress distributions, as shown in Figure 21. Areas with higher cooling efficiency presented a state of residual compressive stress, whilst areas with lower cooling efficiency presented a state of residual tensile stress. In AJC discs, the cooling rate of the central hole surface was much higher than that of other areas of the disc. Owing to the deformation coordination caused by different cooling rates, a radial gradually decreasing residual compressive stress was generated near the surface of the central hole. After complete heat treatment, the value of residual stress still reached 259 MPa, and the range of residual compressive stress reached about 6 mm, whilst the maximum residual tensile stress was 51 MPa outsider 6 mm from the inner hole surface. Considering the low cooling rate, the residual stress of the discs after AC can be ignored. Residual compressive stress also existed in the discs after water cooling, but the value and range were smaller than those in the discs after AJC. The maximum residual compressive stress reached 109 MPa within 3 mm from the surface of the central hole, whilst the maximum residual tensile stress was 37 MPa outsider 3 mm from the inner hole surface.

A comparison of the crack growth life of discs is shown in Figure 22. When the maximum speed of the test was 47,500 rpm, the crack propagation life of the disc cooled by air jetting was 10.7% higher than that of the disc cooled naturally in the air. The crack propagation life of the disc cooled by air jetting was 23.7% higher than that of the water-cooled disc. When the maximum speed of the test dropped to 45,000 rpm, the crack propagation life of the disc cooled by air jetting was 35.9% higher than that of the water-cooled disc. Some conclusions can be drawn from the experimental result: The residual compressive stress near the central hole increased the crack propagation life, and the residual tensile stress reduced the crack propagation life. Therefore, in the subsequent heat treatment scheme, the residual tensile stress at a position of large stress should be avoided.

From Figure 20 and Figure 21, conclusions can be drawn that in the area where residual compressive stress existed, the crack propagated relatively slowly, indicating that the residual compressive stress could effectively inhibit the crack propagation. This inhibition effect was more obvious when the crack size was short. When the crack propagation path passed through the residual tensile stress area, the residual tensile stress in the disc would significantly promote the crack propagation. From the result of external surface morphology detection of crack, with failure and fracture, the crack sizes of differently treated discs were the same. This was likely because the crack passed the residual stress existence area when the disc failed, and the stress fields at the crack front of different discs were similar. In the subsequent test, the crack length and the amplitude of the stress intensity factor can be considered comprehensively to predict the failure life of discs.

To predict the crack propagation behaviour effectively, a classic LEEF approach [41] was used in this work to predict the fatigue life of the discs:(10)$$\frac{da}{dN}=C{\left(\Delta K_{eff}\right)}^{n},\;\Delta K_{th}<\Delta K_{eff}\;<\;K_{Q}$$
where *C* is a material constant, $\Delta K_{eff}$ is the range of the effective stress intensity factor, and $\Delta K_{eff}$ is the sum of $\Delta K_{ext}$ and $\Delta K_{res}.\;\Delta K_{effth}$ is defined as the stress intensity factor threshold value. When $\Delta K_{eff}\;$Δ$\Delta K_{effth}$, the crack grows, and when $\Delta K_{eff}\;$Δ *K_Q_*, it means the disc fails. The parameters for the mean line were found to be *C* = 4.2869 × 10^−14^ mm/cycle, *n* = 3.26 with ΔK expressed in $\mathrm{MPa}\sqrt{\mathrm{mm}}$. Though the residual stresses and the local mean stress varied during crack propagation, it was hard to measure the variation of residual stress at the crack tip. And during the crack propagation, the ratio of residual stress at the crack tip to the maximum stress gradually tended to zero. Therefore, the stress ratio *r* = 0 was considered in this paper.

The crack propagation analysis software FRANC3D and submodel analysis techniques with ABAQUS were used in this work. FRANC3D uses the submodel technology for the repartition of the mesh within the local submodel, thus greatly improving the efficiency rate. Generally, the size of the structural parts to be analysed is relatively large, the cracks are relatively small, and the mesh needs to be very fine. Thus, a local model is needed to qualify the parts containing cracks so that the analysis is considerably focused on the cracks themselves. The calculation flow of crack propagation is shown in Figure 23.

In this paper, the commercial finite element software ABAQUS was used to divide the sub-model and calculate the global model. The sub-model already divided in ABAQUS was imported to FRANC3D to introduce the crack and enlarge it. The sub-model, after introducing the crack, was resubmitted to ABAQUS to calculate the stress field at the crack front under the combined action of centrifugal force and residual stress. FRAC3D would extract the stress and displacement fields at the crack front and calculate the stress intensity factor at the crack front through M-Integral. Then crack propagation was carried out by setting the parameters and step size of crack growth. In this paper, the maximum tensile stress (MTS) theory was used to compute the crack kink angle. It meant that the crack kinked in the direction where tensile stress was ahead of the crack front point. The number of propagation cycles corresponding to the crack length for five different disks after each step was recorded in Figure 24. The calculated residual stress was applied to the crack surface using a 2D stress field method. The positive value represents the residual tensile stress, and the negative value represents the residual compressive stress. And the curve crack propagation rate da/dN with crack length in the double logarithmic coordinate system is drawn in Figure 25. Two conclusions can be drawn from Figure 24. The variation trend of crack growth rate (da/dN) with crack length (a) does not change under the difference of residual stress field, though at the same maximum fatigue speed, the comparison of the crack growth rate at the same crack length is AJC > AC > WQ The crack growth life calculated through finite element simulation of D01–D05 was 3479, 3062, 2855, 3854, and 3077 cycles, respectively. A comparative analysis between the simulation results and the test results of disc crack propagation life is presented in Table 6. This linear elastic finite element analysis with superposition of stress intensity factors is simple and effective only in terms of crack propagation life prediction.

The crack propagation life of the five discs predicted using FRANC3D and replica measurement is shown in Figure 26 and Figure 27. The crack growth life calculated by regarding the residual stress as $\Delta K_{ree}\;$ in FRANC3D was close to the LCF test result. In the initial and later stages of crack growth, differences existed between the crack growth rate obtained by FRANC3D simulation and the fracture extrapolation results. Nevertheless, in the stable crack growth period, the crack growth rate obtained by FRANC3D simulation was consistent with the fracture extrapolation results. The use of linear elastic fracture mechanics enabled calculating crack propagation by linear superposition of residual stress as driving or restraining force on external force.

A conclusion can be drawn from Figure 26 and Figure 27; that is, the process of crack propagation described by FRANC3D simulation and replica measurement exhibited a high degree of coincidence. Hence, taking the linear superposition of rotational load and residual stress as the driving force of crack propagation is reasonable. When the maximum fatigue speed was 47,500 rpm, the crack growth curve from FRANC3D was highly consistent with the replica results. Nonetheless, when the maximum load was 45,000 rpm, the crack growth curve from FRANC3D was significantly slower than the replica results. The reason for this phenomenon was that the residual stress would relax gradually as the test went on, and the degree of stress relaxation would increase with the increase in fatigue load [42,43]; the effect of residual stress on crack growth at high speed was also lower than that at low speed. In general, residual compressive stress could effectively inhibit crack growth by promoting crack closure, whereas residual tensile stress would make the crack tip easier to open and promote crack propagation. From comparing the fracture morphology of D01, D02 and D03 discs in Figure 28, the interface between the crack propagation area and the instantaneous fracture area of the three discs was obvious. The fatigue source area was black due to oxidation, the propagation area was smooth because of the mutual friction of the crack surface, and pits could be observed in the instantaneous fracture area. After different heat treatments, the size of the outer surface of the crack did not change, but great differences in the size of the inner surface of the crack were found. The reason for this phenomenon may be that a certain crack closure effect occurred on the outer surface of the crack. The residual compressive stress in the AJC disc inhibited the opening of the crack surface during crack propagation, whereas the tensile residual compressive stress in the WQ disc promoted the opening of the crack surface during crack propagation.

## 5. Conclusions

In this study, the AJC method was utilised to induce residual compressive stress near the central hole of the GH4169 disc and compared with the AC and WQ methods. An LCF crack propagation test of five discs was conducted using an HR6DI rotor high-speed rotating tester. The main conclusions can be drawn as follows:(1)Different cooling methods in solid solution were adopted to determine the residual stress distribution near the central hole of the discs. Amongst the methods mentioned above, the AJC method could induce about 259 MPa residual compression stress within the range of 6 mm from the surface of the central hole. WQ method results in a smaller range and value of residual compressive stress. In the area 3 mm outside the central hole surface, the residual stress mainly exists in the form of tensile stress. Almost no residual stress occurred near the central hole of the disc after AC because of its low cooling rate. The comparison showed that the AJC method effectively induces appropriate residual compressive stress.(2)The LCF crack propagation test of five prefabricated discs with initial cracks was carried out on the HR6DI rotor high-speed rotating tester. The crack growth life of D01 (AJC at 47,500 rpm), D02 (AC at 47,500 rpm), D03 (WQ at 47,500 rpm), D04 (AJC at 45,000 rpm) and D05 (WQ at 47,500 rpm) was 3497, 3160, 2827, 3888 and 2860, respectively. When the maximum speed of the test was 47,500 rpm, the crack propagation life of the disc cooled by air jetting was 10.7% higher than that of the disc cooled naturally in the air. The crack propagation life of the disc cooled by air jetting was 23.7% higher than that of the water-cooled disc. When the maximum speed of the test dropped to 45,000 rpm, the crack propagation life of the disc cooled by air jetting was 35.9% higher than that of the water-cooled disc. The import of residual compressive stress field near the central hole by AJC effectively inhibited crack propagation. When the fatigue load was low, this inhibition effect was more significant.(3)By taking the linear superposition of residual stress and centrifugal force as the driving force of crack propagation, the crack propagation life of different wheel discs was analysed and calculated using FRANC3D. The results agreed with the results of surface replica and fractured back extrapolation, which proved the feasibility and accuracy of this method. Comparison of the surface replica and fracture back extrapolation results of different heat treatment discs at 47,500 and 45,000 rpm showed that the mechanism of residual stress affecting crack propagation was that compressive stress would inhibit the opening of the crack surface during crack propagation to prevent the initiation and propagation of the crack, whereas residual tensile stress would make the opening of the crack surface easier to have the opposite effect. Therefore, the influence of residual tensile stress should be avoided in material processing.

## Figures and Tables

**Figure 1 materials-15-05228-f001:**
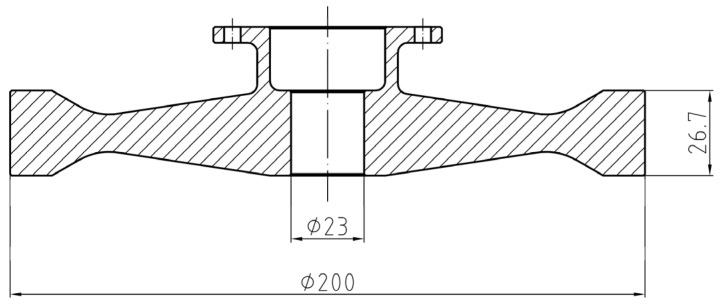
Designed disc geometry and dimensions.

**Figure 2 materials-15-05228-f002:**
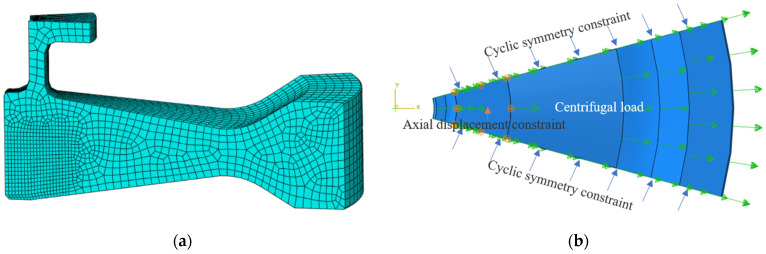
Stress analysis of the disc with the finite element method: (**a**) 3D model and (**b**) load and boundary conditions.

**Figure 3 materials-15-05228-f003:**
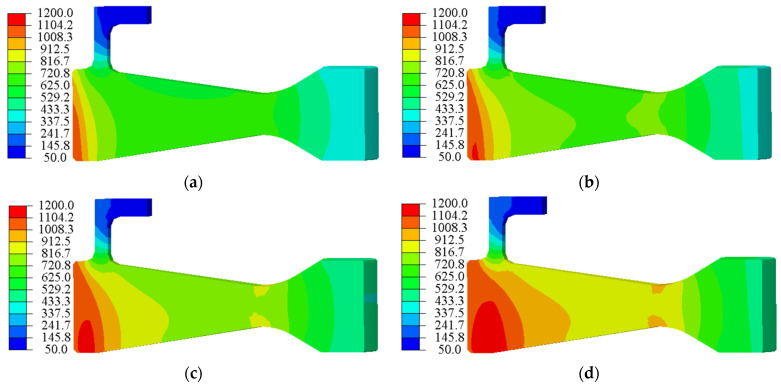
Hoop stress distribution at maximum rotation speed: (**a**) 42,500 rpm, (**b**) 45,000 rpm, (**c**) 47,500 rpm and (**d**) 50,000 rpm.

**Figure 4 materials-15-05228-f004:**
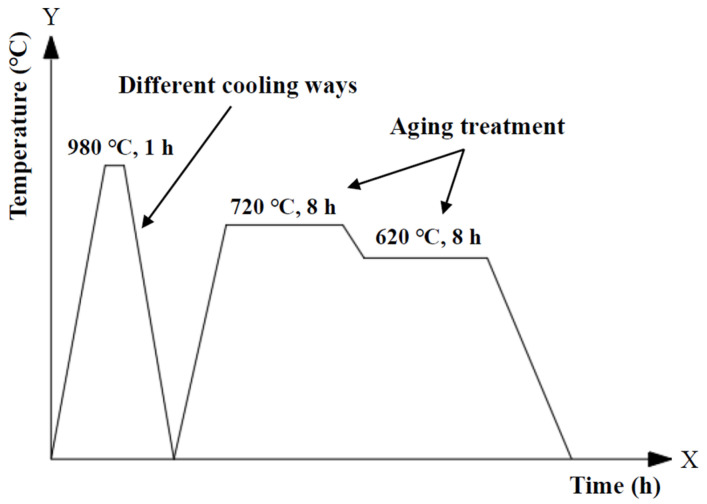
GH4169 alloy heat treatment process.

**Figure 5 materials-15-05228-f005:**
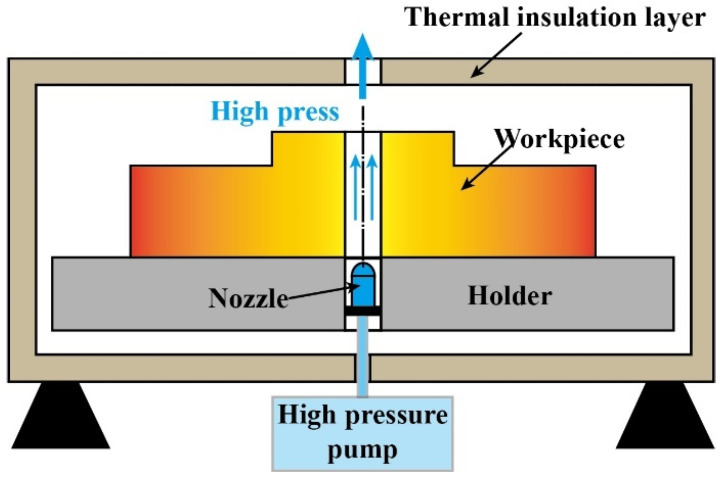
AJC method.

**Figure 6 materials-15-05228-f006:**
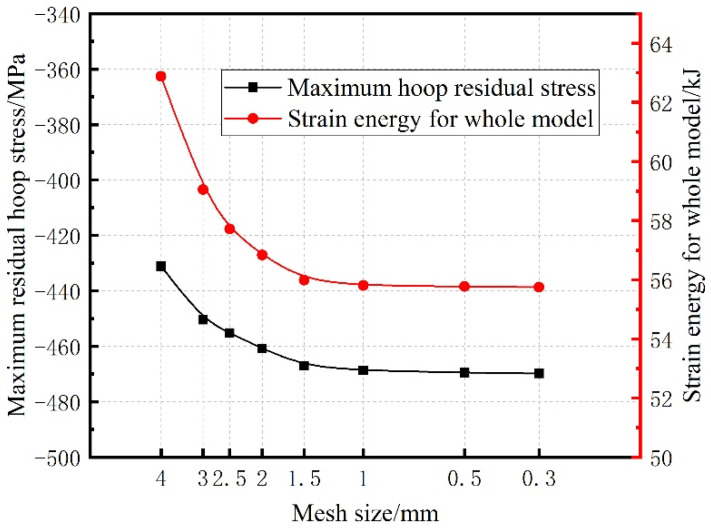
The maximum residual hoop stress and the strain energy of the whole model of the WQ method.

**Figure 7 materials-15-05228-f007:**
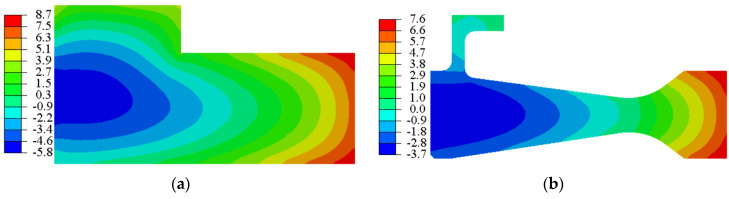
Residual hoop stress after AC. (**a**) Rough machined workpiece (**b**) Finishing disc.

**Figure 8 materials-15-05228-f008:**
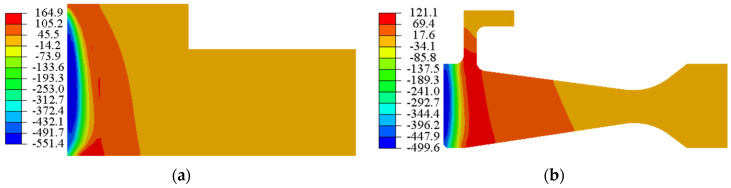
Residual hoop stress after AJC. (**a**) Rough machined workpiece (**b**) Finishing disc.

**Figure 9 materials-15-05228-f009:**
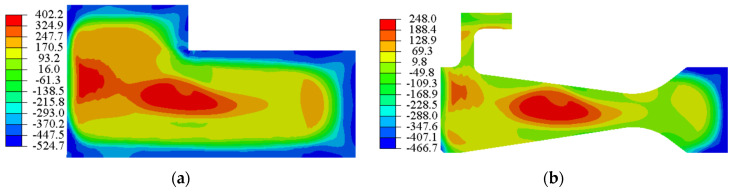
Residual hoop stress after WQ. (**a**) Rough machined workpiece (**b**) Finishing disc.

**Figure 10 materials-15-05228-f010:**
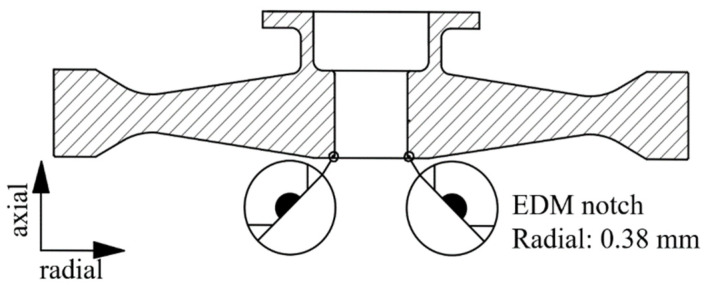
Initial flaws prefabricated by EDM.

**Figure 11 materials-15-05228-f011:**
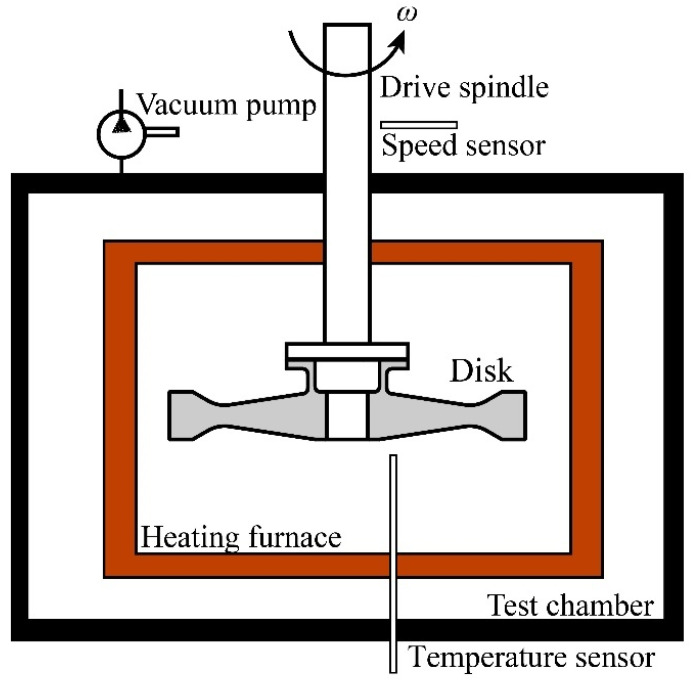
Schematic of the test device.

**Figure 12 materials-15-05228-f012:**
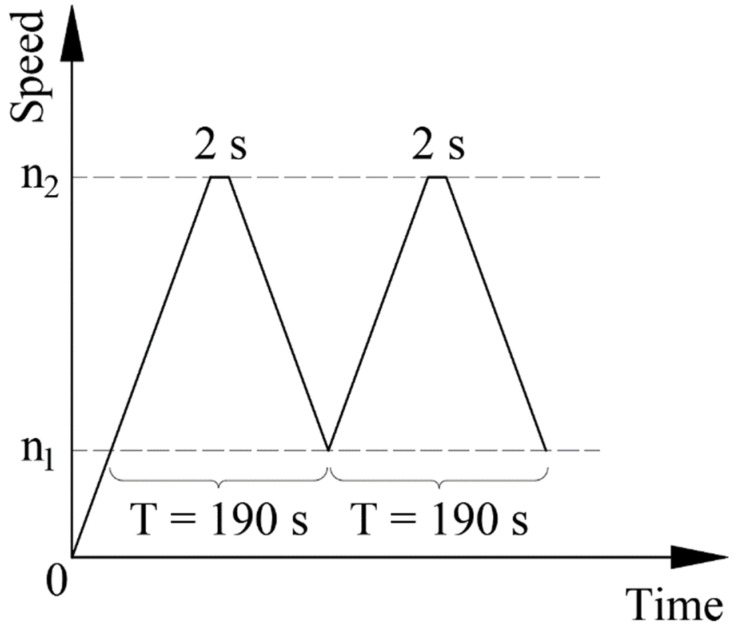
Test load spectrum.

**Figure 13 materials-15-05228-f013:**
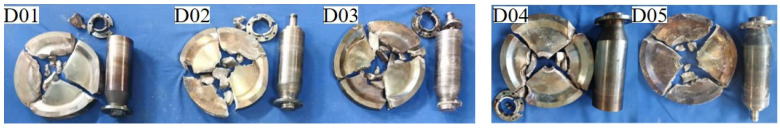
Discs after fatigue failure (D01–D05 are arranged from left to right).

**Figure 14 materials-15-05228-f014:**
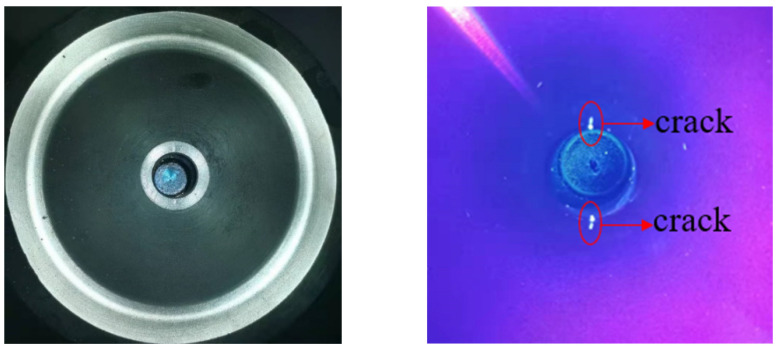
Crack length from fluorescence detection.

**Figure 15 materials-15-05228-f015:**
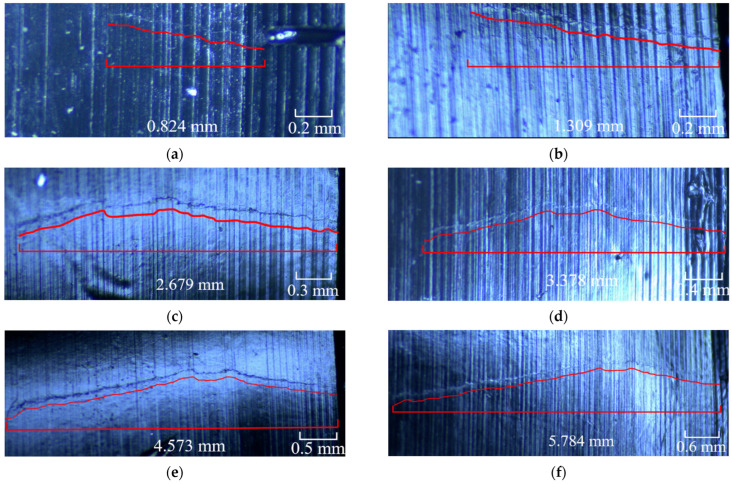
Detecting crack length of D01 disc using the surface replica method. (**a**) 1500 cycles, (**b**) 2000 cycles, (**c**) 2500 cycles, (**d**) 2700 cycles, (**e**) 2900 cycles, (**f**) 3100 cycles, (**g**) 3300 cycles.

**Figure 16 materials-15-05228-f016:**
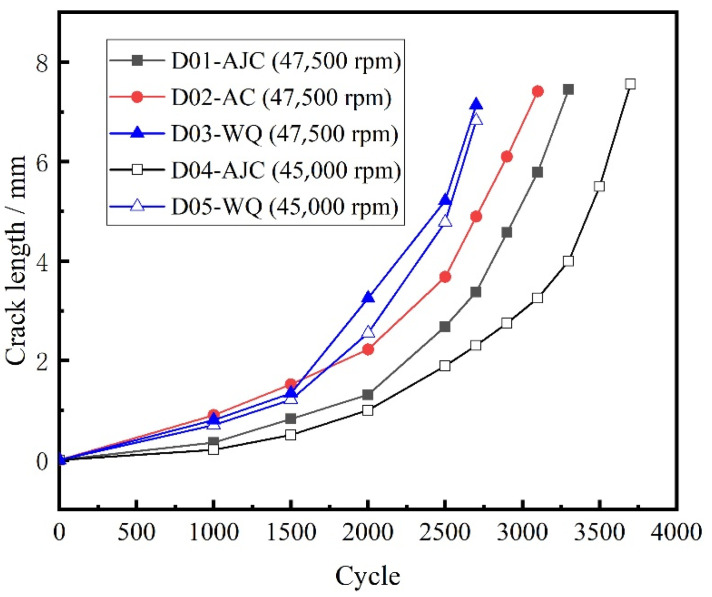
External surface crack length recorded using the surface replica method.

**Figure 17 materials-15-05228-f017:**
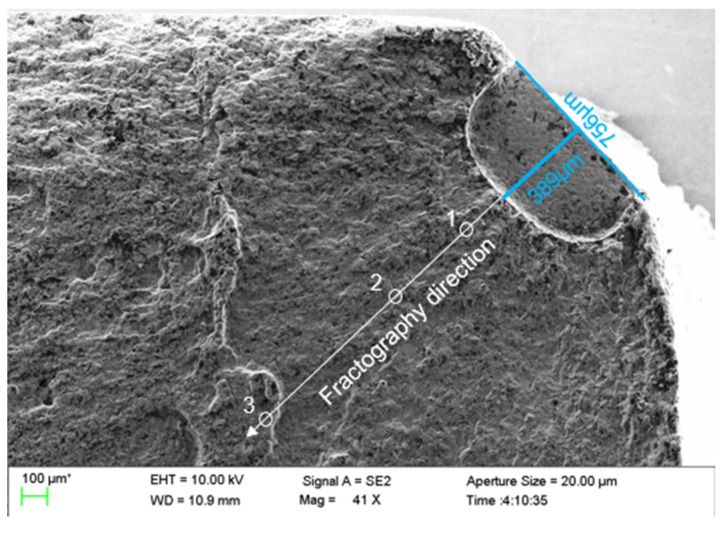
Typical optical image of a fracture surface for a crack emanating from an EDM notch.

**Figure 18 materials-15-05228-f018:**
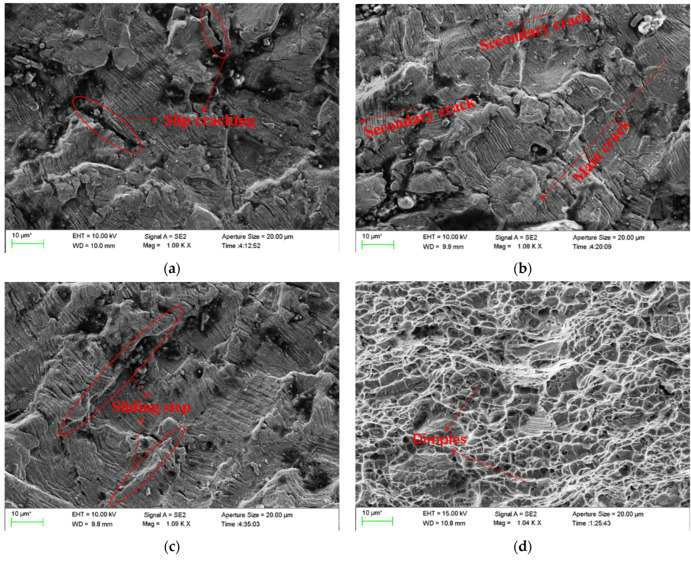
Fracture morphology of the crack in different propagation stages: (**a**–**c**) correspond to locations 1–3 in Figure 17, respectively, and (**d**) represents the dimple morphology of the instantaneous fracture zone.

**Figure 19 materials-15-05228-f019:**
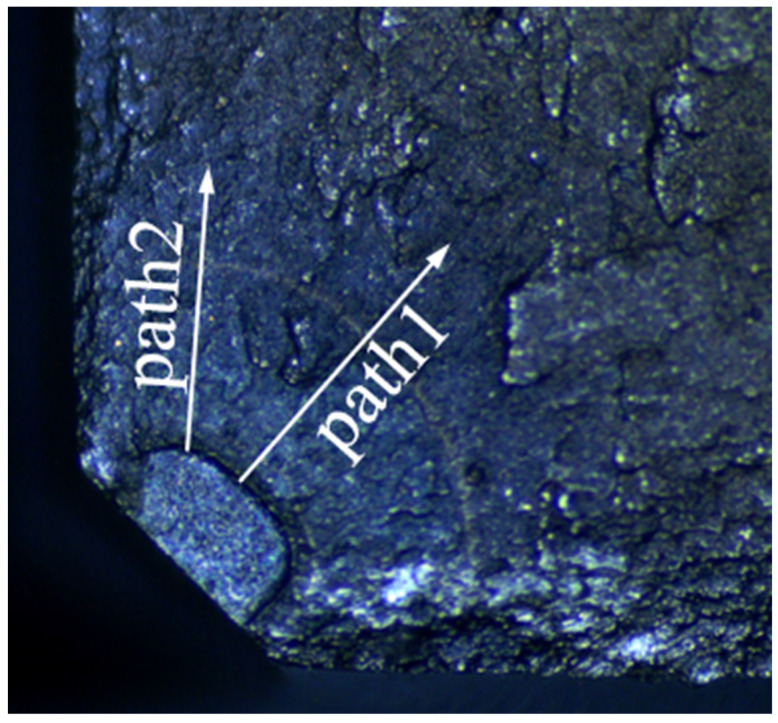
Two paths to measure the crack propagation rate.

**Figure 20 materials-15-05228-f020:**
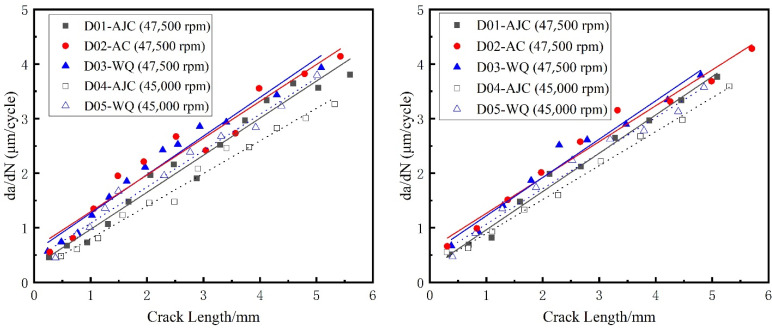
Inner surface crack propagation rate vs. crack length. (Path 1 on the **left**, path 2 on the **right**).

**Figure 21 materials-15-05228-f021:**
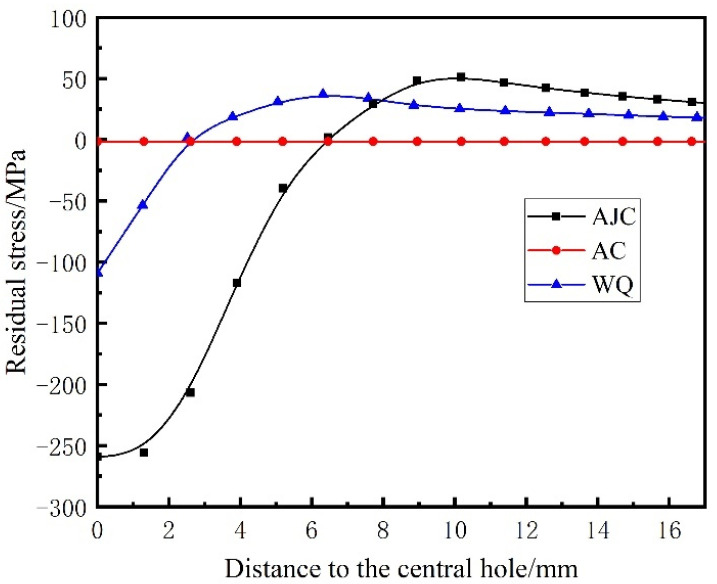
Different residual stress distributions of three cooling methods.

**Figure 22 materials-15-05228-f022:**
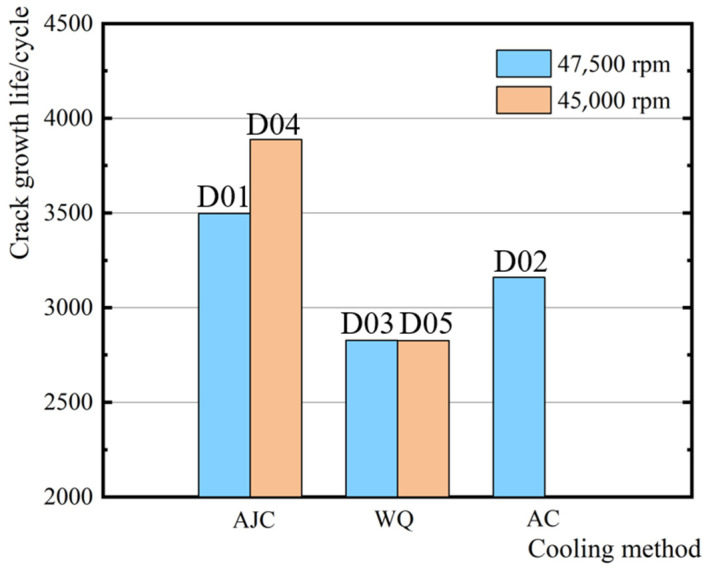
Crack growth life of discs with different cooling methods.

**Figure 23 materials-15-05228-f023:**
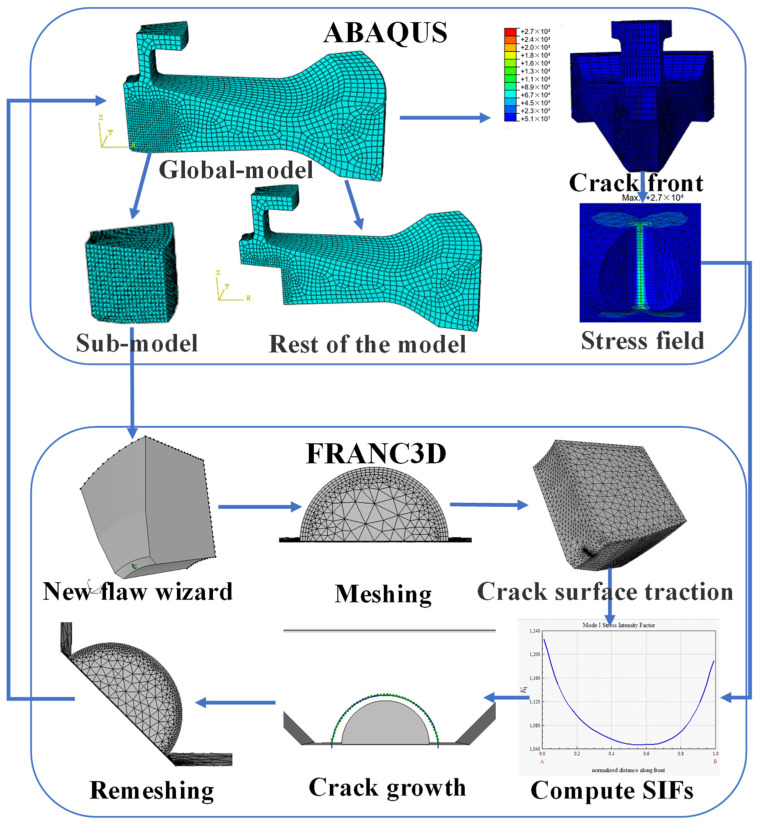
Calculation of crack growth rate by using ABAQUS and FRANC3D.

**Figure 24 materials-15-05228-f024:**
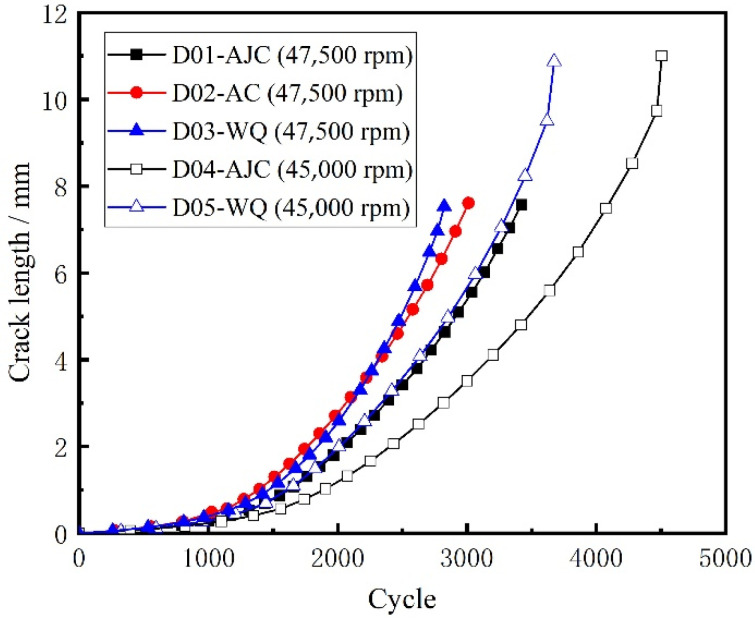
Crack length with cycle by using FRANC3D.

**Figure 25 materials-15-05228-f025:**
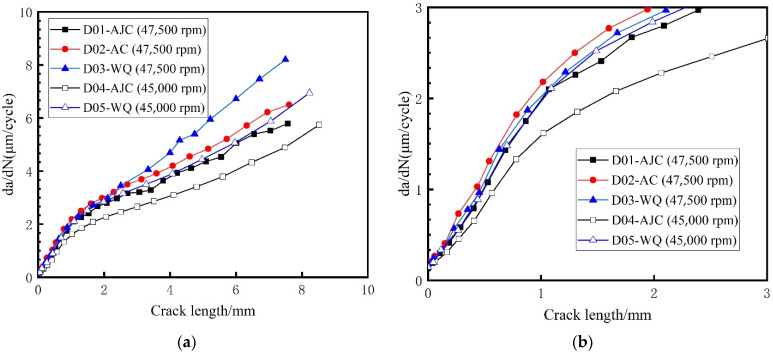
Crack propagation rate da/dN with crack length by using FRANC3D. (**a**) Crack length ranged from 0–10 mm; (**b**) Crack length ranged from 0–3 mm.

**Figure 26 materials-15-05228-f026:**
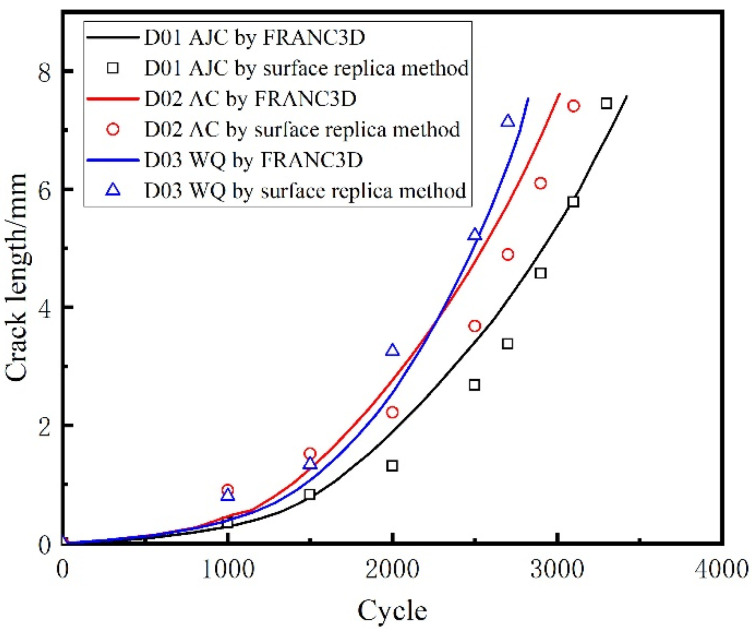
Comparison of crack length between the replica method and FRANC3D simulation at 47,500 rpm.

**Figure 27 materials-15-05228-f027:**
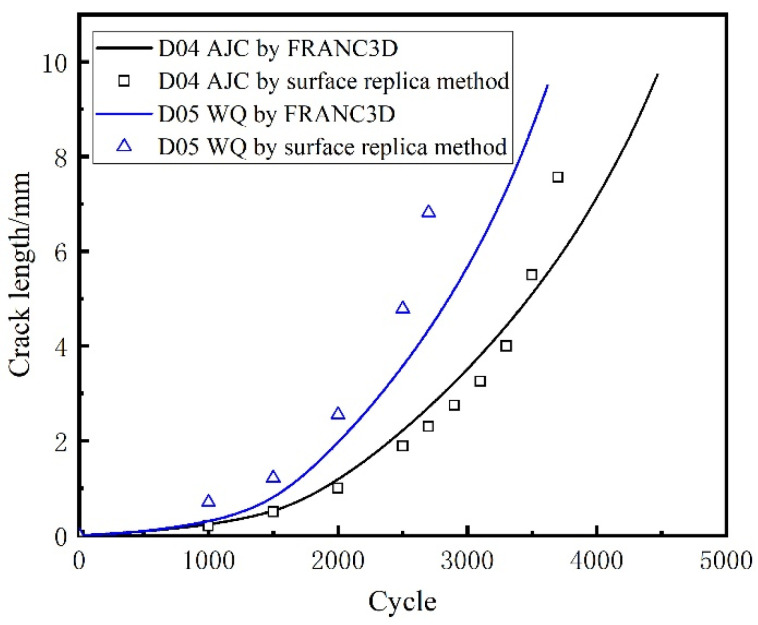
Comparison of crack length between the replica method and FRANC3D simulation at 45,000 rpm.

**Figure 28 materials-15-05228-f028:**
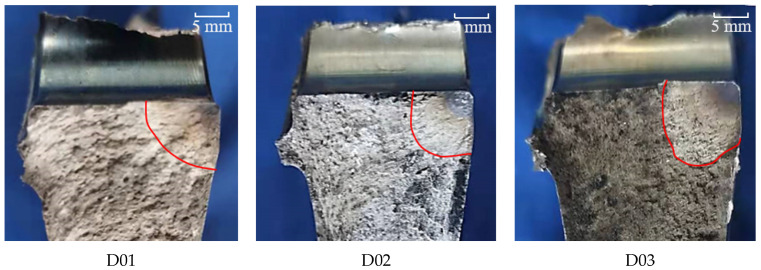
Fracture morphology of D01, D02 and D03.

**Table 1 materials-15-05228-t001:** Mechanical properties of the GH4169 alloy at 500 °C.

	GH4169 (500 °C)
Yield stress σ_0.2_ (MPa)	1050
Ultimate tensile strength σ_b_ (MPa)	1260
Poisson’s ratio ν	0.33
Young’s modulus E (GPa)	175
$\mathrm{Stress}\;\mathrm{intensity}\;\mathrm{factor}\;\mathrm{threshold}\;\Delta K_{\mathrm{th}}\;(\mathrm{MPa}\sqrt{m}$)	18
$\mathrm{Conditional}\;\mathrm{fracture}\;\mathrm{toughness}\;KQ_{\;}(\mathrm{MPa}\sqrt{m}$)	101.3

**Table 2 materials-15-05228-t002:** Relationship between thermodynamic parameters and temperature.

Temperature °C	Thermal Conductivity W/(m °C)	Density $10^{3}kg/m^{3}$	Specific Heat Capacity J/(kg/°C)	Coefficient of Thermal Expansion 10^−6^	Thermal Convective Coefficient $J/\left(m^{2}s\;{}^{\circ}C\right)$	Young’s Modulus GPa
0	11.02	8226	424	12.8	4000	197
100	12.75	8190	434	13.1	4500	197
200	14.36	8160	448	13.4	4400	197
300	15.96	8130	463	13.8	4050	197
400	17.60	8090	480	14.2	3660	197
500	19.18	8050	500	14.0	3330	196
600	20.77	8010	525	15.1	2880	194
700	22.36	7960	560	15.7	2700	187
800	23.95	7910	605	16.4	2520	165
850	24.53	7890	625	16.8	-	145
900	25.10	7860	636	17.1	1800	130
1000	26.83	7810	645	17.5	1000	105

**Table 3 materials-15-05228-t003:** Designed discs.

	Cooling Method	AJC	AC	WQ
Max Speed				
47,500 rpm		D01	D02	D03
45,000 rpm		D04	-	D05

**Table 4 materials-15-05228-t004:** Experimental details of disc crack propagation.

Test Parameter	Value
Disk temperature	500 ± 5 °C
Chamber vacuum degree	<100 Pa
Maximum rotation speed	47,500/45,000 rpm
Minimum rotation speed	3150 rpm
Disk acceleration	400 r/(min·s)

**Table 5 materials-15-05228-t005:** Results of disc crack propagation test.

Disk Num.	Crack Growth Life	Max Rotation Speed	Cooling Method
D01	3497	47,500	AJC
D02	3160	47,500	AC
D03	2827	47,500	WQ
D04	3888	45,000	AJC
D05	2860	45,000	WQ

**Table 6 materials-15-05228-t006:** Comparative analysis between simulation and test results.

	Test Life	Predicted Life	Error Percentage
D01-AJC method (maximum speed is 47,500 rpm)	3497	3421	−2.17%
D02-AC method (maximum speed is 47,500 rpm)	3160	3012	−4.68%
D03-WQ method (maximum speed is 47,500 rpm)	2827	2823	−0.14%
D04-AJC method (maximum speed is 45,000 rpm)	3888	4506	15.9%
D05-WQ method (maximum speed is 45,000 rpm)	2860	3674	28.5%

## Data Availability

Not applicable.

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
