# Peer review of "Effect of Residual Stress Induced by Different Cooling Methods in Heat Treatment on the Fatigue Crack Propagation Behaviour of GH4169 Disc"

_materials, 2022, doi:10.3390/ma15155228_

Round 1
Reviewer 1 Report
Review of manuscript #1827147
The manuscript describes a computational and experimental study of the influence of residual stress on crack growth in GH4196 discs following three different heat treatment protocols.
The research topic is interesting and will interest the readers of Materials. The general research methodology is good, incorporating numerical, analytical and experimental methods including fracture characterization. Nevertheless, there are several major questions and topics that must be addressed prior to publication of the manuscript.
Major Issues:
1. The authors base their assessment of the residual stress on the numerical solution. Unfortunately, the authors do not show or address the issue of solution verification. It must be demonstrated that the computed residual stress fields are not affected by the choice of mesh size. Convergence plots for the numerical solution in both local field values such as stress or strain and global solution values such as strain energy should be shown.
2. In section 2.3 the authors describe the governing equations for the different cooling methods investigated. Only the heat transfer equations are addressed. It is not clear how the thermomechanical coupling between the temperature and the displacement field was addressed? The authors should greatly elaborate on this topic.
3. In Figure 8 the stress distribution in the rough work-piece and the disc are shown. It is not clear how the material removal from the rough work-piece to the finished product was taken into account? Were element simply removed from the computed solution?
4. The computed residual stress distribution is not validated in any way in the study. I would assume a simple hardness test (indention) at various locations on the disk surface, following heat treatment, could at the list validate the residual stress distribution on the specimen outer boundary. With no verification (see Issue 1) and no validation the authors cannot demonstrate that the assumed residual stress distribution is valid.
5. In figure 18 the crack propagation is shown as a function of crack size for the different cases investigated. Since only a very limited number of experiments were conducted the authors must address the issue of the small experimental sample size. It may be that if more experiments were conducted the differences shown in crack propagation would vanish or become more prominent.
6. What crack propagation criterion was used in the FRANC3D code? How was the crack propagated? In which direction? The whole process of crack propagation simulation and data extraction used to construct figure 22 should be made more clear to the readers.
Minor issues:
1. More information regarding the numerical models should be given. What program was used to compute the residual stress fields? What element type? What solution scheme?
2. In Table 2, why is the thermal convection coefficient dependent on the temperature? Please explain.
3. It is not clear what the authors want to show in Figure 17. It should be better explained and marked on the Figure.
4. Figure 21 is very hard to read. This Figure should be greatly enlarged so the different legends be readable.
Reviewer 2 Report
The article highlights peculiarities of the effects of residual stress induced by three different cooling methods during heat treatment on the crack propagation behaviour of the GH4169 disc. The authors successfully combined simulation and test investigation. However, it would be good to experimentally determine the level of residual stresses after different modes of heat treatment.
The article is interesting, but a number of shortcomings need to be corrected:
1. References to articles by other authors should be added in the introduction, in which they evaluate the influence of the residual stress level on fatigue crack growth, for example https://doi.org/10.1007/s11003-013-9539-9.
2. The font size in Figs. 4, 5, 9, 10, 15, 18, 19, 20, 21, 22, 23, 24 and 25 should be increased.
3. The resolution of Fig. 26 is low, and the scale bar should be added.
4. More new References (2019-2022) should be added.
Reviewer 3 Report
Materials-1827147
The paper presents and experimental and numerical analysis of the effect of residual stresses on low-cycle fatigue life of nickel-based superalloy discs. Different residual stress states have been induced by means of different cooling means after solution treatment. Three cooling means have been adopted, namely air cooling, air jet cooling and water quenching. Aging treatment following the solution treatment was the same in all three cases. Residual stresses were computed by means of transient thermal simulations, since microstructural changes affecting the final residual stress distribution were not involved. After evaluating the residual stress distributions on rough forgings, the effect of final machining on the residual stress distribution was simulated by the death of finite element, according to my understanding, but this step is unclear. Low cycle fatigue tests have subsequently been executed by inserting a small pre-crack using the EDM and by monitoring the crack propagation using the replica method. FRANC 3D simulations have been run by considering the driving force consisting of the applied SIF (owing to the rotational speed) and the residual SIF (owing to the residual stress field). Eventually a comparison between experimental and numerical crack growth data has been done.
In my opinion the paper report an interesting application, but the following amendments are necessary to improve the paper clarity:
11) Introduction: “The crack growth date was recorded by fluorescence…” should be “The crack growth data was recorded by fluorescence…”
22) Figure 2: some details regarding the FE model are missing:
a. Which FE software package has been used? Which element type? How many degrees of freedom has the model?
b. Why didn’t the Authors use an axisymmetric 2D model?
33) Table 1: what load ratio does DKth refer to?
44) Figure 3: I suggest that all four figures are plotted with reference to the same contour scale for making the comparison more simple.
55) Section 2.3: “The thermal convective coefficient of high-pressure jetting air was 1800 J/(m2s·°C), whereas it was only 20 J/(m2s·°C) in stable air”. How did the Authors determine the coefficient for convective heat transfer? Please clarify
66) Section 2.3: “Lastly, the stress changes from the rough machined workpiece to the finishing disc were calculated through the setting of the life and death unit.” The concept conveyed by this statement is obscure: please expand and explain it more clearly.
77) Figure 11: please report the time values on the x-axis: how long does it take to complete a load cycle?
88) Caption to figure 17: shouldn’t Fig 16 be cited?
99) Caption to figure 19: “three” should be used instead of “there”
110) Section 4: “When the crack propagation path passed through the residual tensile stress area, the residual compressive stress in the disc would significantly promote the crack propagation.”. Please check the meaning and the construction of the sentence: the residual COMPRESSIVE stresses should REDUCE (and not PROMOTE) crack propagation
111) Section 4: “The parameters for mean line were found to be C=4.2869×10-14 mm/cycle,n=3.26 with ΔK expressed in MPa sqrt mm .”.
a. How did the Authors determine the material parameters of the Paris law?
b. It is well known that the material parameters of the Paris law depend on the load ratio (or, alternatively stated, the mean stress): in the course of crack propagation, the residual stresses which the crack tip undergoes vary, therefore the local mean stress also vary: how did the Authors take into account the variation of the local stress ration (or local mean stress) during crack propagation? This point must be discussed in the revised version of the paper
112) Section 4: “A conclusion can be drawn from Figure 23”. Shouldn’t it be Figure 24? Please check
113) Figure 25: the figure shows that FRANC3D predicts a slower crack propagation rate than measured experimentally. Unfortunately, the explanation provided by the Authors: “The reason for this phenomenon was that the residual stress would relax gradually as the test went on, and the degree of stress relaxation would increase with the increase in fatigue load [33–34]; the effect of residual stress on crack growth at high speed was also lower than that at low speed.” Why FRANC3D simulation did not capture the stress relaxation? This point must be clarified.
114) Conclusions: the circumstance that the residual stresses have not been measured but they have been estimated by means of transient FE thermal simulations must be recalled.
Round 2
Reviewer 1 Report
The authors have addressed all my questions and comments.
I recommend publication of the revised manuscript.
Reviewer 2 Report
The authors took into account all comments of the reviewer and made appropriate corrections to the manuscript.